# *Capnocytophaga canimorsus* Endocarditis Presenting with Leukocytoclastic Vasculitis and Glomerulonephritis

**DOI:** 10.3390/microorganisms12102054

**Published:** 2024-10-12

**Authors:** Divya Chandramohan, Nilam J. Soni, Moyosore Awobajo-Otesanya, Terrence Stilson, Min Ji Son, Ariel Vinas, Rushit Kanakia, Riya N. Soni, Marcos I. Restrepo, Gregory M. Anstead

**Affiliations:** 1Division of Infectious Diseases, Department of Medicine, University of Texas Health San Antonio, 7703 Floyd Curl Drive, San Antonio, TX 78229, USA; anstead@uthscsa.edu; 2Division of Pulmonary Diseases & Critical Care, Department of Medicine, University of Texas Health San Antonio, 7703 Floyd Curl Drive, San Antonio, TX 78229, USArestrepom@uthscsa.edu (M.I.R.); 3Medicine Service, South Texas Veterans Health Care System, 7400 Merton Minter Blvd, San Antonio, TX 78229, USA; riyasoni0802@gmail.com; 4Department of Pathology, University of Texas Southwestern Medical Center, 5323 Harry Hines Blvd., Dallas, TX 75390, USA; moyosore.awobajo-otesanya@utsouthwestern.edu; 5Joe R. Teresa Lozano Long School of Medicine, University of Texas Health San Antonio, 7703 Floyd Curl Drive, MC 7885, San Antonio, TX 78229, USA; stilson@livemail.uthscsa.edu (T.S.); sonm1@uthscsa.edu (M.J.S.); 6Division of Cardiology, South Texas Veterans Health Care System, 7400 Merton Minter Blvd, San Antonio, TX 78229, USA; vinas@uthscsa.edu (A.V.); kanakiaright@gmail.com (R.K.)

**Keywords:** *Capnocytophaga*, endocarditis, tricuspid valve, glomerulonephritis, dog bite, vasculitis

## Abstract

*Capnocytophaga canimorsus* is a gram-negative bacterium commonly found in the saliva of dogs and cats. Despite the frequency of animal bites, infection with *Capnocytophaga* species is rare, and severe infections are usually associated with underlying risk factors, such as alcohol use disorder, asplenia, or immunosuppression. We describe a case of a man who presented with a purpuric rash, lower extremity edema, and acute renal failure and was found to have tricuspid valve endocarditis and infection-associated glomerulonephritis due to *C. canimorsus*. Despite treatment with cefepime, the vegetation increased in size and valvular function worsened. He was readmitted with an inferior wall myocardial infarction, heart failure, and pulmonary embolism. He underwent an urgent tricuspid valve replacement with a bioprosthetic valve. A 16S ribosomal RNA amplicon sequencing performed on the resected valve tissue verified involvement of *C. canimorsus*. Post-operatively, he had several episodes of gastrointestinal hemorrhage requiring multiple endoscopic interventions and arterial embolization. The recurrent gastrointestinal hemorrhage combined with his severe functional decline ultimately led to his death. This patient had an uncommon presentation with leukocytoclastic vasculitis and infection-associated glomerulonephritis, which revealed an underlying diagnosis of infective endocarditis due to *C. canimorsus,* a rare gram-negative bacterial etiology of infective endocarditis.

## 1. Introduction

*Capnocytophaga canimorsus* is a gram-negative bacterium commonly found in the saliva of dogs and less commonly, in cats. Despite the frequency of dog bites, infections with this organism are rare, however, serious infections can occur, including bacteremia, meningitis, endophthalmitis, and endocarditis, usually in patients with underlying risk factors, including alcohol use disorder, asplenia, and immunosuppression [1].

We describe the case of a man with a history of alcohol abuse who presented with a purpuric rash, lower extremity edema, and acute renal failure, and was found to have *C. canimorsus* tricuspid valve (TV) endocarditis and glomerulonephritis. Endocarditis is an uncommon manifestation of *C. canimorsus* infection [2], and only two other cases of *Capnocytophaga* spp. causing endocarditis with glomerulonephritis have been reported [3,4]. This case report highlights the varied manifestations of *C. canimorsus* infection and demonstrates how systemic *C. canimorsus* infection can result in severe morbidity.

## 2. Case Presentation

A 67-year-old male presented with a one-month history of worsening bilateral lower extremity rash and edema. He first noticed a non-pruritic, petechial, and purpuric rash around his ankles that ascended to his thighs, abdomen, and wrists (Figure 1).

He subsequently developed bilateral ankle edema that progressed to 2+ bilateral lower extremity edema to the knees. Three days prior to presentation, he developed bilateral knee pain that limited ambulation and prompted him to seek medical care. He also reported oliguria. He denied fever, cough, headache, diarrhea, and abdominal or chest pain.

His medical history was notable for hypertension, prediabetes, and tobacco and alcohol use disorders. He reported currently drinking two to three beers and smoking one pack of cigarettes daily. He was retired and lived with his wife. He denied any recent travel or current illicit drug use.

Physical examination was remarkable for 2+ bilateral lower extremity pitting edema to the shins and non-palpable petechiae and purpura extending from his legs to his abdomen and from his upper arms to his wrists. Healed scars from prior dog bites were noted on both hands.

Initial laboratory studies indicated an elevated white blood cell count of 15.8 K/µL (reference range (RR) 4–10 K/µL), 85% neutrophils (RR 44–75%), lymphocytes 8% (RR 16–44%), platelet count of 403 K/µL (RR 150–400 K/µL), albumin level of 2.7 g/dL (RR 3.5–5.7 g/dL), and an elevated creatinine level of 2.9 mg/dL (RR 0.7–1.3 mg/dL). He had a significant anemia, with a hemoglobin of 8.2 g/dL (RR 13.5–17.5 g/dL) and a mean corpuscular volume of 87.7 fL (RR 78–98 fL). (His hemoglobin level was normal at 17.8 mg/dL six months prior to admission.) His serum iron level was 42 µg/dL (RR 50–212 µg/dL), total iron binding capacity 219 µg/dL (RR 225–400 µg/dL), transferrin 188 mg/dL (RR 203–362 mg/dL), and ferritin 539 ng/mL (RR 10–322 ng/mL), signifying anemia of chronic disease without iron overload. A urinalysis showed a protein level >500 mg/dL, 396 white blood cells per high powered field (hpf), 1450 red blood cells/hpf, and no eosinophils. The urine protein/creatinine ratio was 6.9 (RR 0–0.2). A urine drug screen was positive for marijuana. The high-sensitivity C-reactive protein level was 10.2 mg/dL (RR 0–1.0 mg/dL). The patient’s chest X-ray was unremarkable, and bilateral lower extremity Doppler ultrasound studies were negative for deep venous thromboses.

His clinical presentation was concerning for a small-vessel vasculitis with renal involvement, and he was admitted to the hospital to expedite diagnostic evaluation and initiate hemodialysis. Subsequent rheumatologic tests that were within normal limits included anti-cyclic citrullinated peptide, anti-ribonucleoprotein, anti-Smith, antinuclear, atypical p-antineutrophilic cytoplasmic, perinuclear, and cytoplasmic antibodies. Rheumatoid factor was elevated at 44.9 IU/mL (RR < 14 IU/mL). Serum complement levels were low (C3 < 15 mg/dL (RR 87–200 mg/dL) and C4 of 8 mg/dL (RR 19–52 mg/dL)). Cryoglobulins were positive at 1%, with an IgM monoclonal protein with kappa light chain specificity. A polymerase chain reaction assay [Biofire FilmArray BCID2 panel (detects 25 species of bacteria, and seven species of yeast)] performed on the blood culture specimen was negative. Renal ultrasound showed normal-sized kidneys and increased cortical echogenicity consistent with medical renal disease. A timed urine protein measurement was 3550 mg/24-h.

The purpuric rash was clinically consistent with leukocytoclastic vasculitis per dermatologic consultation, although a skin biopsy was not performed. A transthoracic echocardiogram showed normal ejection fraction, but revealed a large tricuspid valve (TV) vegetation (Appendix A). A transesophageal echocardiogram confirmed large, multilobed vegetations on the TV measuring 1.9 cm × 0.9 cm on the anterior leaflet and 2.2 cm × 0.7 cm on the posterior leaflet associated with severe tricuspid regurgitation (Figure 2, Appendix A). (For a video of the transesophageal echocardiogram with cardiologist commentary, see Appendix A.)

Two sets of blood cultures grew gram-negative rods after 48 h, which were identified as *C. canimorsus* (Figure 3). Blood cultures cleared after three days of antibiotic treatment. Upon further questioning, the patient recalled having a dog bite while playing with his dog three months prior. He thought the wound was likely infected, but did not seek medical attention and let the wound heal on its own.

Given his worsening renal function, a renal biopsy was performed and showed diffuse acute proliferative glomerulonephritis (markedly hypercellular glomeruli with neutrophilic infiltrates, interstitial acute and chronic inflammatory infiltrate, and endothelial cell swelling) consistent with an impression of glomerulonephritis (Figure 4). Hemodialysis was initiated as his renal function worsened.

The constellation of findings confirmed a diagnosis of *C. canimorsus* infective endocarditis of the TV with leukocytoclastic vasculitis and acute proliferative glomerulonephritis. The patient was started on piperacillin-tazobactam and transitioned to intravenous ceftriaxone 2 g three times per week with hemodialysis. Given the size of the vegetation (>2 cm), severity of tricuspid regurgitation, and persistent lower extremity edema, TV replacement was planned during his hospitalization. However, the patient desired to go home and was discharged against medical advice with plans to complete six weeks of antibiotic therapy with intravenous cefepime and follow up with cardiothoracic surgery as an outpatient. Cefepime was chosen due to its availability in his outpatient hemodialysis center.

Before outpatient surgery could be performed, he was readmitted to the hospital with dyspnea, bradycardia, and hypotension. A computed tomography angiogram of the chest revealed right upper and lower lobe pulmonary emboli (Figure 5), and intravenous heparin was started. A transesophageal echocardiogram showed growth of the TV vegetation to 4.0 × 3.0 cm (Appendix A). A transthoracic echocardiogram showed newly reduced left ventricular systolic function (ejection fraction of 35–40%); a dilated, hypokinetic right ventricle; and the absence of an intracardiac shunt. (For a video of the transesophageal echocardiogram with cardiologist commentary, see Appendix A.)

An electrocardiogram showed intermittent complete heart block and a troponin I level that was persistently elevated (1.40 ng/mL; RR < 0.08 ng/mL). Left heart catheterization revealed total occlusion of the mid-right coronary artery (Figure 6, Appendix A), and a drug-eluting stent was placed requiring dual antiplatelet therapy with resolution of the heart block. Dual antiplatelet therapy with cangrelor and aspirin was started.

The patient was restarted on piperacillin-tazobactam and was prepared for an urgent TV replacement. Intraoperatively, there was a 3 cm vegetation on the anterior leaflet of the TV, with >50% of the leaflet destroyed (Figure 7). He had a successful TV replacement with a bioprosthetic valve with a residual paravalvular leak (Figure 8, Appendix A). His blood and valve cultures were negative. However, 16S ribosomal RNA amplicon sequencing (University of Washington, Seattle, WA, USA) performed on the resected valve tissue was consistent with *C. canimorsus* infection. Antibiotic treatment was continued with piperacillin-tazobactam.

The patient was transitioned to cangrelor, given his recent stent during the perioperative period, and it was held prior to surgery as per protocol. Unfortunately, his postoperative course was complicated by hemopericardium requiring surgical drainage (Appendix A) and multiple episodes of gastrointestinal bleeding requiring blood products, esophagogastroduodenoscopy (EGD), and coiling of the gastroduodenal artery by interventional radiology. He was discharged to a skilled nursing facility after a prolonged hospital stay. One week later, he was readmitted with melena and underwent EGD with successful hemostasis of the bleeding source. However, the patient continued having episodic gastrointestinal hemorrhage requiring vasopressor support. Due to his recurrent bleeding and poor functional status, the patient and his family decided to transition to comfort care. The patient died two and half days later, which was 112 days after his initial presentation with lower extremity rash.

## 3. Discussion

Approximately 80% of cases of infective endocarditis are due to *Staphylococcus, Streptococcus,* and *Enterococcus* species, particularly *Staphylococcus aureus* and the viridans group of streptococci [5]. Among gram-negative bacteria, the most common pathogens causing infective endocarditis are *Escherichia coli* and *Klebsiella* species. *Captocytophaga canimorsus* is a rare cause of infective endocarditis, occurring in <2% of cases of bacteremia due to this organism [6].

Based on our literature review using MEDLINE, a least 31 cases of *C. canimorsus* endocarditis have been reported (Table 1). Hino and Veltman [2] reviewed 25 cases of *C. canimorsus* endocarditis from 1977 to 2021, and since then an additional six cases, including our own, have been reported (Table 2).

Microbiological Characteristics. The genus *Capnocytophaga* includes nine different species of gram-negative bacilli that are facultative anaerobes. *Capnocytophaga canimorsus* grows on blood agar in 1 to 14 days, with an average incubation period of 4 to 5 days, and its slow growth may delay identification [1]. 16S Ribosomal sequencing can diagnose and differentiate *C. cynodegmi* and *C. canimorsus* when blood cultures are negative [11]. Only the oxidase-positive strains of *C. cynodegmi* cause severe infections, which would explain the extremely low prevalence of *C. cynodegmi* infections, despite it constituting a larger proportion of the dog and cat oral microbiome in comparison with *C. canimorsus* [12,13]. The virulence-associated capsular polysaccharides A, B, and C have been found in a minority of the strains of *C. canimorsus* and *C. cynodegmi* populating the dog oral flora, explaining the overall low incidence of severe infections secondary to *Capnocytophaga* spp. One study found that 92% of human *C. canimorsus* infections are caused by serovars A, B, and C, but these three serovars constitute only 8% of the isolates of *C. canimorsus* found in dog saliva [14].

Transmission. *C. canimorsus* is found in dog and cat saliva and is transmitted to humans by animal bites, scratches, or contact. A dog bite predisposes to *C. canimorsus* infection, as does close canine contact [1,11,15,16], although cases from cat bites, including a lion, and feline contact, have been reported [17]. Our patient had a long history of bites and scratches from playing with his dog, with a recent bite three months prior to the onset of illness.

Risk Factors. Multiple risk factors predispose to *C. canimorsus* infection. An underlying cardiac condition was the most common risk factor predisposing an individual to endocarditis from *C. canimorsus*, and was seen in 39% of cases, followed by alcohol use disorder, which was seen in 27% [Table 2]. Alcohol consumption impairs neutrophil function and can worsen severity of *C. canimorsus* infection, along with iron overload; however, absence of these risk factors makes severe infections unlikely given the low virulence of this organism [1]. Impairment of host defenses, including asplenia (either anatomic or functional), and immunosuppression (glucocorticoid use or cancer chemotherapy) are other important risk factors. No identifiable predisposing condition was found in 41% of the individuals in one series [1]. Janda et al. stated that 70% of patients in their study of *C. canimorsus* infections were >50 years of age, with men representing 57% of cases [11], which were additional risk factors of our patient. Male patients afflicted with *C. canimorsus* endocarditis were younger in our literature review (median of 51 years versus 69 years for females) [Table 2]. The source of infection was most commonly secondary to a dog bite in 43%, or contact with a dog in an additional 36% of cases [Table 2]. Uncommon modes of acquisition of *C. canimorsus* bacteremia were from dog scratches, exposure to dog feces, and a lion bite, and no animal contact was apparent in 7% of cases [Table 2]. Our patient had a history of alcohol use disorder, which is also an endocarditis risk factor, although the specific mechanism of alcohol use being implicated in endocarditis has not been elucidated [18].

Injection of foreign antigens via drug use results in antibody production and immune-complex deposition on valve surfaces, and these areas serve as sites for bacterial adhesion [19]. Intravenous drug use is known to damage cardiac valves particularly the TV with repetitive assault from particulate matter in the injected drug [20]. Our patient, however, did not have a history of injection drug use.

Clinical Manifestations. Human infections due to *Capnocytophaga* are rare, but serious manifestations can occur, including bacteremia, meningitis, ocular infections, and endocarditis. Initial symptoms of *C. canimorsus* infections are non-specific, including fever, malaise, myalgia, vomiting, diarrhea, abdominal pain, dyspnea, confusion, headache, and skin manifestations [11,21]. A retrospective review of 31 cases of *Capnocytophaga* spp. endocarditis demonstrated that 97% of patients presented with fever [16]; however, our patient did not have fever. Instead, our patient presented with lower extremity purpuric rash and edema, and he denied any constitutional symptoms. *Capnocytophaga canimorsus* infections are commonly associated with a rash. Purpura is seen in 37%, often with petechiae and/or ecchymoses, in addition, a macular or maculopapular rash is seen in 13% [1]. Eschars developed at the sites of dog bites in two asplenic patients presenting with *C. canimorsus* sepsis and disseminated intravascular coagulopathy [22].

Even in the setting of *C. canimorsus* bacteremia, endocarditis is rare. Sandoe noted only 12 cases from 1977–2002, occurring in a 4.5:1 male-to-female ratio, of which 25% died [23]. In a rabbit model, a large inoculum of *C. canimorsus* was necessary to cause endocarditis due to the low virulence of the organism, and treatment of rabbits with methylprednisolone prolonged the duration of the bacteremia and predisposed them to endocarditis [24]. Interestingly, splenectomy, which prolongs the duration of bacteremia, does not appear to increase the incidence of endocarditis [24]. This is also in concordance with Sandoe’s review, in which none of the patients with endocarditis had a history of asplenia [23].

The predominant native valve affected with *C. canimorsus* endocarditis is the aortic valve (60.7%), followed by the TV (39.3%) [see Table 2 for details]. The most common complications of *C. canimorsus* endocarditis are heart block in 10% of cases, and valvular abscess, congestive cardiac failure, and atrioventricular shunt in 6% of cases each [Table 2]. Other cardiac complications include myocardial infarction, aortic root pseudoaneurysm, and aortic regurgitation [Table 2]. Embolic phenomena were also observed in 16% of patients, with pulmonary embolism being the most common (10% of all cases). Valve surgery was undertaken in 63% of cases [Table 2].

Endocarditis presenting with cutaneous vasculitis has been reported previously but is uncommon [25]. In a retrospective review of 766 cases of cutaneous vasculitis, only 3.5% (n = 27) of cases were associated with a bacterial infection and 0.8% (n = 6) had endocarditis [26]. Another immunological phenomenon is infection-associated glomerular injury causing membranoproliferative and crescentic glomerulonephritis, which has been reported in three cases of *Capnocytophaga* spp. endocarditis [3,4], including our patient. The immunopathogenesis of glomerulonephritis is different in our case as compared to the prior cases reported by Archer and by Kannan and Mattoo [3,4]. Although all three cases had glomerular deposition of complement, our patient and Kannan and Mattoo’s patient had evidence of IgA, IgM, and IgG deposition. Only our patient had reported cryoglobulinemia. Additionally, our patient presented with a higher creatinine level that worsened quickly and was the only case that required hemodialysis. The three cases of endocarditis due to *Capnocytophaga* species and infection-associated glomerulonephritis are compared in Table 3.

Medical Therapy. Intravenous antibiotics are the primary treatment for endocarditis caused by *C. canimorsus*. Typically, the initial antibiotic regimen includes a β-lactam/β-lactamase inhibitor combination or carbapenem because there has been increasing prevalence of β-lactamase producing strains of *Capnocytophaga* species, although β-lactamase production by *C. canimorsus* is uncommon [27,28,29,30]. *Capnocytophaga canimorsus* is often susceptible to a variety of antibiotics including penicillin, cephalosporins, carbapenems, macrolides, quinolones, and rifampin [23]. Various regimens have been reported to successfully treat *C. canimorsus*, including penicillin in 39%, cephalosporins in 39%, aminopenicillins in 29%, carbapenems in 19%, and antipseudomonal penicillins in 10% [16].

Surgical Therapy. In recent years, earlier surgical management for bacterial endocarditis has been recommended to reduce the risk of septic embolism [31]. The European Society of Cardiology and the American College of Cardiology/American Heart Association have slight differences in guidelines for the surgical treatment of infective endocarditis. Both guidelines consider heart failure, uncontrolled infection, and embolism risk as surgical indications. However, the timing of surgery (emergent/urgent/elective vs. early), the definition of large vegetation (>3 cm vs. >1 cm), and the combination of vegetation size and embolism as indications for surgery differ between the European and American guidelines. These guidelines focus on left-sided infective endocarditis, and only the European guidelines provide specific recommendations for surgical treatment of right-sided endocarditis: (1) bacteremia despite appropriate antibiotics, (2) persistent large TV vegetation >2 cm and pulmonary emboli despite appropriate antibiotics, and (3) heart failure despite aggressive treatment [32]. Most importantly, multidisciplinary endocarditis management teams are now recommended which have been shown to improve outcomes, including reduction in mortality and hospital length of stay [31].

Six proposed indications for surgical intervention of right-sided infective endocarditis by Shmueli et al. in 2020 [33] include:(1)Microorganisms difficult to eradicate (e.g., persistent fungi);(2)Persistent bacteremia for >7 days despite adequate antimicrobial therapy;(3)Large, persistent TV vegetations (>20 mm);(4)Recurrent pulmonary emboli with or without concomitant right heart failure;(5)Right heart failure secondary to severe tricuspid regurgitation;(6)Cardiac abscess (more common in the setting of an infected prosthetic valve).

A majority (65%) of *Capnocytophaga* spp. endocarditis cases ultimately required surgical intervention along with antibiotic therapy [16]. Our patient declined valve replacement during his initial hospitalization and wanted to postpone surgery to an elective outpatient procedure. His TV vegetations increased in size despite receiving six weeks of intravenous cefepime, most likely due to the poor penetration of cefepime into the large vegetations due to the inoculum effect [34]. In this case, delaying surgical intervention led to a complicated clinical course that ultimately resulted in the patient’s death.

Mortality. A review of 31 cases of *Capnocytophaga* species endocarditis that included five cases of non-*Capnocytophaga canimorsus* species found that treatment with a cephalosporins and embolic phenomena were associated with increased mortality [16]. An early review of 12 patients with *C. canimorsus* infections in France reported an overall mortality of 30% [1]; however, based on our review of 31 *C. canimorsus* endocarditis cases reported to date (Table 2), the overall mortality is approximately 16%.

In summary, *C. canimorsus* is now recognized as an emerging zoonotic pathogen due to widespread dog and cat ownership and increasing numbers of older and immunocompromised persons. Although *Pasteurella* species is recognized as the most common cause of infection related to dog bites and contact with canine saliva [35], *Capnocytophaga* can cause life-threatening complications, including septic shock, gangrene of the digits or extremities, bacteremia, meningitis, endocarditis, and glomerulonephritis. Although uncommon, clinicians should be aware that infective endocarditis can present with cutaneous vasculitis, and initiation of immunosuppression without an infectious work-up or delays in initiation of antibiotic therapy can be detrimental in these patients. Furthermore, clinicians should consider domestic pet exposure a risk factor for atypical infections and unusual disease presentations, and pet owners must be counseled that bacteria carried by healthy pets can potentially cause serious infections, especially in immunocompromised patients.

## Figures and Tables

**Figure 1 microorganisms-12-02054-f001:**
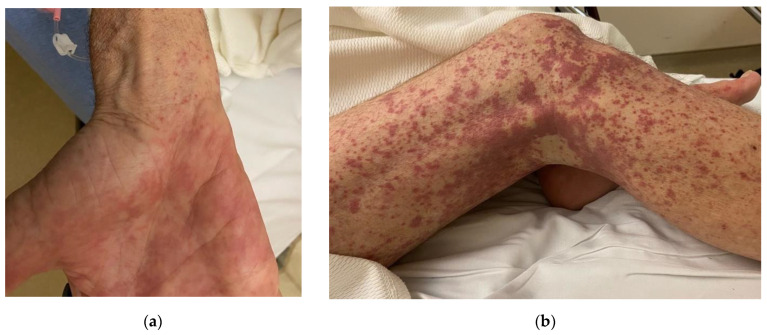
Rash associated with *C. canimorsus* bacteremia and endocarditis. Purpuric rash of (**a**) the right palm and (**b**) the right lower extremity. The left lower extremity was similar, but the rash was less pronounced.

**Figure 2 microorganisms-12-02054-f002:**
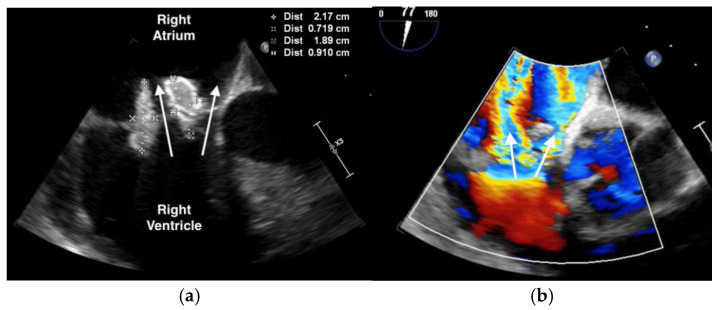
Transesophageal echocardiogram showing: (**a**) large, multi-lobed tricuspid valve vegetations associated with valve perforation and (**b**) severe tricuspid valve regurgitation (white arrow).

**Figure 3 microorganisms-12-02054-f003:**
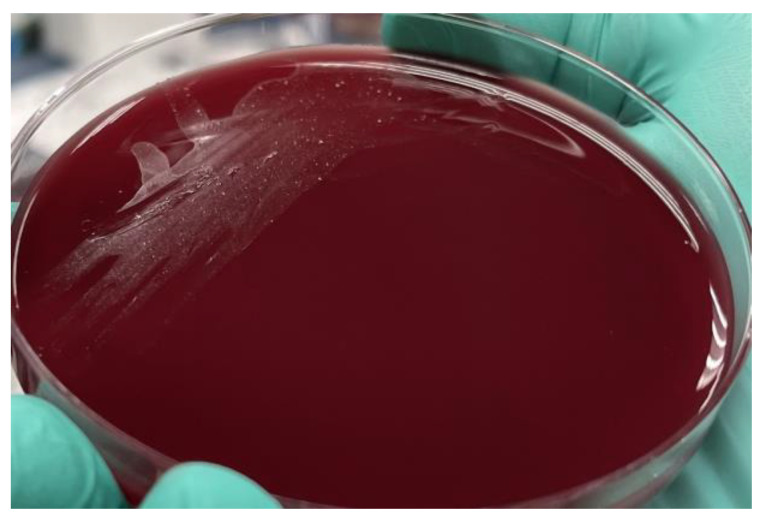
Characteristic pinpoint colonies of *C. canimorsus,* as seen on blood agar.

**Figure 4 microorganisms-12-02054-f004:**
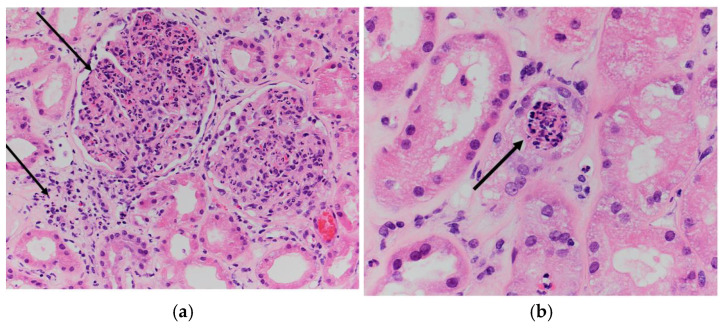
Renal biopsy showing diffuse acute proliferative glomerulonephritis: (**a**) global glomerular enlargement and hypercellularity with dense neutrophilic and histiocytic infiltrate, as well as interstitial acute inflammation and edema (black arrow) (Hematoxylin and Eosin (H&E), 200× magnification) and (**b**) endothelial cell swelling with capillary luminal obstruction by acute inflammatory cells (black arrow) (H&E, 400× magnification).

**Figure 5 microorganisms-12-02054-f005:**
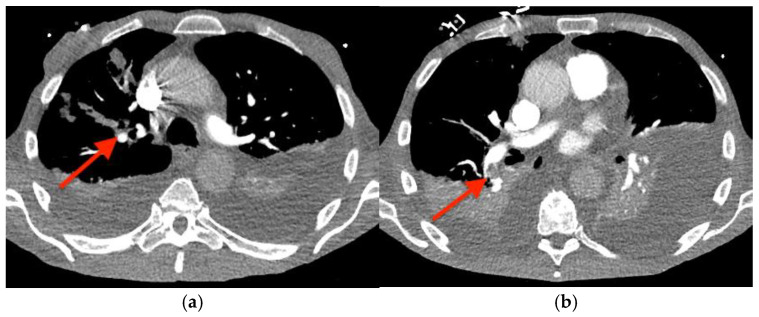
Computed tomography angiogram of the chest showing pulmonary emboli (red arrows) in: (**a**) the right upper lobe and (**b**) the right lower lobe.

**Figure 6 microorganisms-12-02054-f006:**
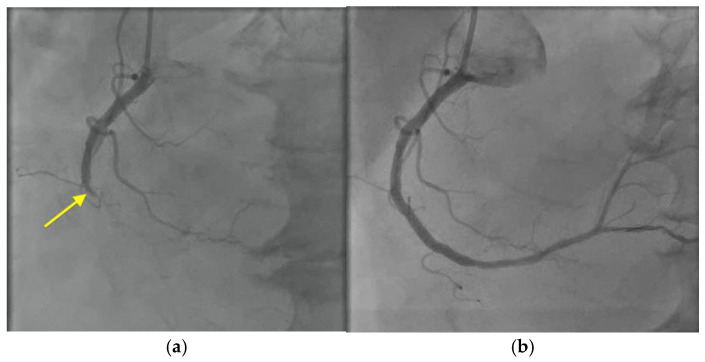
Cardiac catheterization with right coronary artery angiogram showing: (**a**) total occlusion of the right coronary artery (yellow arrow), followed by (**b**) restoration of blood flow after deployment of a drug-eluting stent.

**Figure 7 microorganisms-12-02054-f007:**
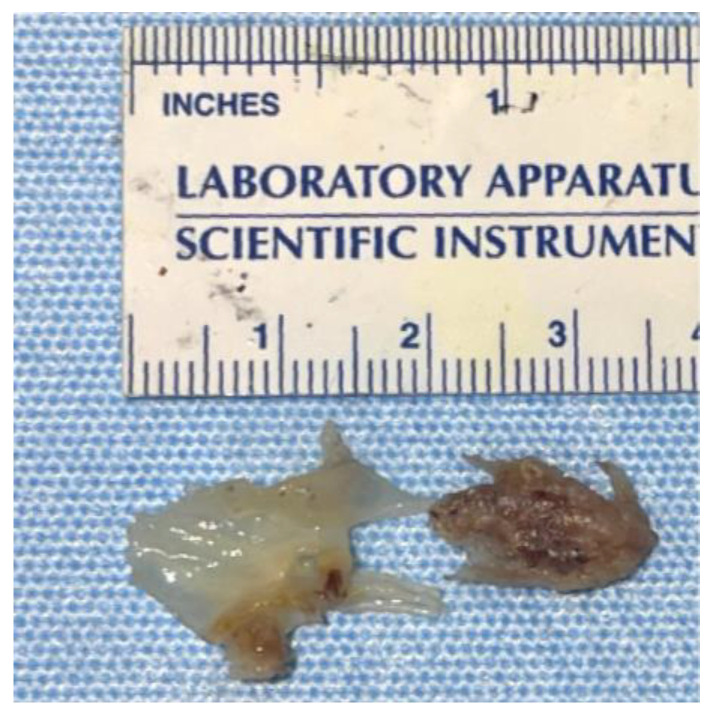
Resected tricuspid valve tissue showing the attached vegetation.

**Figure 8 microorganisms-12-02054-f008:**
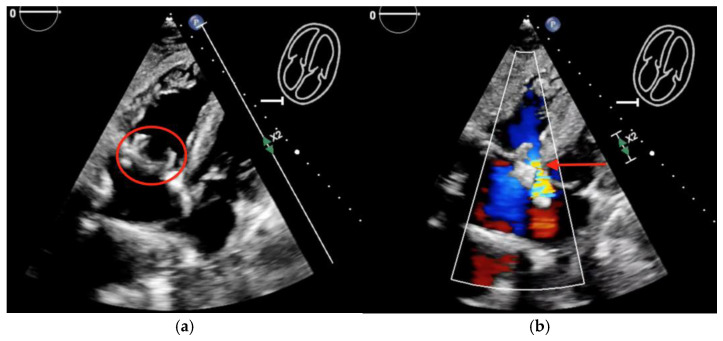
Postoperative transthoracic echocardiogram showing: (**a**) new bioprosthetic tricuspid valve (red circle) and (**b**) residual paravalvular leak (red arrow).

**Table 1 microorganisms-12-02054-t001:** *Capnocytophaga canimorsus* endocarditis cases reported since the 2021 review by Hino and Veltman [2].

Reference, Year	Age, Sex, Comorbidities	Source	Valve(s) Involved	Complications	Surgery	Antibiotics (Duration)	Outcome
Present case	67 M, HTN, tobacco & alcohol use	Dog bite	TV	GN, anemia, PE, vegetation growth, MI, 3rd degree heart block	Yes	Cefepime (6 weeks) followed by pip-tazo	Recovered after surgery, died of unrelated cause
Harrigan [6], 2022	76 M, HTN, DM II, smoking, alcoholism	Dog bite	AV, MV	3rd degree heart block; Atrial- ventricular fistula	Yes	Ceftriaxone (not specified)	Recovered
Hedman [7], 2023	80 F, healthy	Dog bite	AV	Septic shock, acute kidney injury, emboli to toes	No	Pip-tazo (7-days); ampicillin and ciprofloxacin (28 days)	Recovered; mild aortic regurg present 5 weeks after presentation
Salazar-Rodriguez [8], 2023	39 M, healthy	Dog bite	AV	Aortic root pseudoaneurysm; right atrium to left ventricle fistula	Yes	Pip-tazo (not specified); ceftriaxone (8 weeks), concurrent metronidazole (4 weeks)	Resolved
Gonzalez [9], 2023	80 F, HTN, aortic stenosis	Dog bite	AV, MV	T4-T9 epidural abscess; glenohumeral septic arthritis; 3rd degree heart block; aortic peri-annular abscess; stroke	No	Amoxicillin-clavulanate (not specified); ceftriaxone (not specified)	Died due to respiratory failure after stroke
O’Dwyer [10], 2022	33 M, bicuspid aortic valve, anomalous origin of RCA from LCA	Dog bite	AV	Aortic root abscess, aortic regurgitation, acute decompensated heart failure	Yes	Ceftriaxone (6 weeks), gentamicin and vancomycin (2 weeks)	Recovered

Abbreviations: AV, aortic valve; DM II, diabetes mellitus type II; GN, glomerulonephritis; HTN, hypertension; MI, myocardial infarction; MV, mitral valve; PE, pulmonary embolism; pip-tazo, piperacillin-tazobactam; regurg, regurgitation; TV, tricuspid valve; RCA, right coronary artery; LCA, left coronary artery.

**Table 2 microorganisms-12-02054-t002:** Summary of the characteristics of 31 patients with *Capnocytophaga canimorsus*
^a^ endocarditis.

Characteristic	Result
% male (n = 29)	69
Age, yrs median (range)	56 (24–80); Female 69 years (41–80); Male 51 years (24–76)
Comorbidities (n = 26)	Underlying cardiac diagnosis: 10 total; prosthetic valve (2); aortic stenosis (3); murmur (2); atrial myxoma (1); ICD (1); pacemaker (1); atrial fibrillation (1); bicuspid aortic valve (1); anomalous origin of RCA from LCA (1); rheumatic mitral valve disease (1) Hypertension (4); COPD (3); tobacco use (3); underlying cancer (3); type 2 diabetes (2); renal insufficiency (2); steroid use (1); dyslipidemia (1); nephrectomy (1); osteoarthritis (1) Alcohol abuse: 7 IV drug abuse: 1 None: 6
Source (n= 28)	Dog bite: 12 (42.9%) Dog contact: 10 (35.7%) Dog scratch: 1 (7.4%) Dog feces: 1 (3.6%) Known dog exposure: 25/28 (89.3%) Unknown: 2 (7.1%) Lion bite: 1 (3.6%)
Valve(s) (n = 28 native valves)	AV only: 11/28 (39.3%) TV only: 8/28 (28.6%) MV only: 3/28 (10.7%) AV/TV: 3/28 (10.7%) AV/MV: 3/28 (10.7%) 2 prosthetic valves, 1 ICD
Complications	Cardiac: heart block (3); A-V fistula (2); valvular abscess (2); CHF (2); myocardial infarction; aortic root pseudoaneurysm; aortic regurgitation Embolic: pulmonary embolism (3); stroke; toe emboli Renal: glomerulonephritis (2); acute kidney injury (2) Other: anemia (4); septic shock; septic arthritis; epidural abscess
Surgery	19/30 (63.3%)
Outcome, survival	26/31 (83.9%)

^a^ Includes the 25 patients of Hino and Veltman [2] and the six patients of Table 1. Abbreviations: AV, aortic valve; A-V, atrioventricular; ICD, implantable cardiac defibrillator; MV, mitral valve; TV, tricuspid valve; CHF, congestive heart failure.

**Table 3 microorganisms-12-02054-t003:** Characteristics of glomerulonephritis associated with endocarditis due to *Capnocytophaga* species.

Characteristic	Present Case	Archer, 1985 [3]	Kannan, 2001 [4]
Cr at presentation, mg/dL	2.9	1.2	3.2
Urine WBC/hpf	396	Not described	Not described
Urine RBC/hpf	1450	60	20–40
Proteinuria, g/24-h	3.6	3.0	2+ proteinuria
Serum C3 level	Low	Low	Normal
Serum C4 level	Low	Normal	Normal
Cryoglobulins	Present; IgM monoclonal protein with kappa light chain specificity	Absent	Not described
ANA	Absent	Absent	Absent
Light microscopy	Glomeruli show endocapillary proliferation with abundant neutrophil infiltrate; reactive visceral and parietal epithelial cells are present forming minicrescents; Fibrin occ. associated w/the crescents; duplication of the glomerular basement membrane not identified; mild intimal thickening of arteries w/o vasculitis; moderate lymphoplasmacytic and neutrophil infiltrate of interstitium with occ. aggregates of eosinophils; neutrophil infiltrates of tubules w/occ. WBC casts. Rare necrotic tubules associated with WBC casts; moderate interstitial fibrosis	Increased mesangium pushing into capillary loops.	Glomeruli show segmental necrosis and crescents of varying sizes. Heavy infiltrate of inflammatory cells seen in the interstitium. No vascular lesions seen.
Fluorescence microscopy	Glomeruli are granularly stained: IgG 2-3+, IgA 1+, IgM 1+, C3 3+, C1q 1+, Fibrin Negative, Kappa 1+ and Lambda 1+	Negative results for IgG and IgM and weakly positive results for complement.	2+ granular IgM and C3 in a global mesangial and capillary wall distribution, and IgA, IgG, CIq, 1+ granular in a segmental mesangial distribution.
Electron microscopy	Ultrastructural study of parts of two glomeruli shows prominent endocapillary proliferation with numerous neutrophils in glomerular capillary lumen and mesangial matrix; poorly formed immune complex deposits primarily within the mesangial and paramesangial portions of glomeruli. Hump subepithelial deposits not identified; effacement of the overlying visceral epithelial foot processes associated w/ swelling of visceral epithelial cell cytoplasm and occ. poorly formed pseudovillous transformation. Vacuolar degeneration of the tubular epithelia.	Features consistent with type 1 membranoproliferative GN and not with type 2, or “dense deposit” disease	Glomeruli show cellular crescents, one associated w/ the rupture of the capillary wall. Finely granular and microfibrillary deposits seen in mesangium and paramesangium. Interstitium contained an inflammatory cell infiltrate. No vasculitis seen.
Overall impression	Resembles postinfectious GN	Type 1 membranoproliferative GN (decreased serum C3 level, increased mesangial tissue and capillary wall thickening; milder course than type 2); resembles postinfectious GN	Necrotizing and crescentic immune-complex glomerulonephritis.
Outcome	Required hemodialysis; no indication of improvement over 12 weeks at the time of death	Recovered	Recovered

Abbreviations: Cr, creatinine; WBC, white blood cells; RBC, red blood cells; C3, complement 3; C4, complement 4; ANA, anti-nuclear antibody; occ., occasional; w/, with; w/o, without; IgG, immunoglobulin G; IgM, immunoglobulin M; IgA, immunoglobulin A; C1q, complement 1q; GN, glomerulonephritis.

## Data Availability

The data are contained within the article.

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
