# Peer review of "Capnocytophaga canimorsus Endocarditis Presenting with Leukocytoclastic Vasculitis and Glomerulonephritis"

_microorganisms, 2024, doi:10.3390/microorganisms12102054_

Round 1

Reviewer 1 Report

Comments and Suggestions for Authors

Excellent case report that will help add more knowledge this rare and fatal organism.

The authors mentioned several times in their discussion the association of this organism with animal bites including 1 from a lion bite but did not provide the reference, please add it.

Author Response

Reviewer 1 Comments:

Excellent case report that will help add more knowledge this rare and fatal organism.

Comment 1: The authors mentioned several times in their discussion the association of this organism with animal bites including 1 from a lion bite but did not provide the reference, please add it.

Response: Thank you for reviewing our manuscript.  We have added the reference for the case involving a lion bite.

Reviewer 2 Report

Comments and Suggestions for Authors

Dear authors of the work microorganisms-3233919, I have some constructive observations to make about your work.

Lines 41-42 Please use (,) instead of (.), the whole paragraph is the same idea

Line 51 Why do you cite the conclusion of your work? [1]

Figure 3 please use a microscope to show the characteristics of the colonies

Figure 4 shows two microphotographs, both of which were labeled with the letter (b)

The two microphotographs do not have the same dimensions

Please include a dimensional bar

Figure 7 please try to edit the image, the ruler is uneven horizontally

Line 233 please use (,) instead of (.)

Lines 238-241 the idea requires a bibliographic citation

Line 249 the reference [11] is not placed in the correct place, I suggest placing it in (…57% of cases [11], which were …)

Lines 249-252 the ideas require bibliographic citations

Line 253 please change (.) to (,)

Lines 254-256 why do you cite when describing your results? Please indicate how citation 18 relates to your results

Lines 257-259 please omit the period (.) and replace it with (,) to unite both ideas

Line 264 please omit the period (.) and replace it with (,) to unite both ideas

Line 268 the reference [16] is not placed in the correct place, I suggest placing it in (…with fever[16]; however …)

Line 268 please indicate “our patient presented” instead of “he presented”

Lines 270-272 please change (.) to (,) the whole text is the same idea

Lines 276-280 please change (.) to (,) the whole text is the same idea

Lines 284-286 the idea requires a bibliographic citation

Lines 286-287 the idea requires a bibliographic citation

Lines 288-289 the idea requires a bibliographic citation

Line 289 the idea requires a bibliographic citation

Lines 293-295 the reference [3,4] is not placed in the correct place, I suggest placing it in (…endocarditis [3,4], including our patient.)

Lines 358-370 it does not seem to me that the marked text is a summary

Author Response

Reviewer 2 Comments:

Dear authors of the work microorganisms-3233919, I have some constructive observations to make about your work.

Comment 1: Lines 41-42 Please use (,) instead of (.), the whole paragraph is the same idea

Response: Thank you for your comment. We have changed to (,), instead of (.) as advised.

Comment 2: Line 51 Why do you cite the conclusion of your work? [1]

Response:Thank you for your comment. This incorrect citation has been removed.

Comment 3: Figure 3 please use a microscope to show the characteristics of the colonies

Response:Thank you for your comment. The microscopy image showed gram-negative rods that were barely discernible, and hence, were eliminated from the final manuscript. The colonic morphology on the blood agar plate has been magnified in this revised manuscript.

Comment 4: Figure 4 shows two microphotographs, both of which were labeled with the letter (b)

Response: Thank you for your comment. This has been rectified to indicate (a), and (b).

Comment 5: The two microphotographs do not have the same dimensions. Please include a dimensional bar

Response: Thank you for your comment. We have included the magnification index in the legend to clarify the dimensions. We were unable to include a dimensional bar and embed to the image.

Comment 6: Figure 7 please try to edit the image, the ruler is uneven horizontally

Response: Thank you for your comment. This has been changed.

Comment 7: Line 233 please use (,) instead of (.)

Response: Thank you for your comment. This has been changed.

Comment 8: Lines 238-241 the idea requires a bibliographic citation

Response: Thank you for your comment. We have added the table of characteristics which has this information.

Comment 9: Line 249 the reference [11] is not placed in the correct place, I suggest placing it in (…57% of cases [11], which were …)

Response: Thank you for your comment. We have rectified this.

Comment 10: Lines 249-252 the ideas require bibliographic citations

Response: Thank you for your comment. We have added the table of characteristics which has this information.

Comment 11: Line 253 please change (.) to (,)

Response: Thank you for your comment. We have rectified this.

Comment 12: Lines 254-256 why do you cite when describing your results? Please indicate how citation 18 relates to your results

Response: : Thank you for your comment. We have clarified this statement.

Comment 13: Lines 257-259 please omit the period (.) and replace it with (,) to unite both ideas

Response: Thank you for your comment. This has been changed.

Comment 14: Line 264 please omit the period (.) and replace it with (,) to unite both ideas

Response: Thank you for your comment. This has been changed.

Comment 15: Line 268 the reference [16] is not placed in the correct place, I suggest placing it in (…with fever[16]; however …)

Response: Thank you for your comment. This has been changed.

Comment 16: Line 268 please indicate “our patient presented” instead of “he presented”

Response: Thank you for your comment. This has been changed.

Comment 17: Lines 270-272 please change (.) to (,) the whole text is the same idea

Response: Thank you for your comment. This has been changed.

Comment 18: Lines 276-280 please change (.) to (,) the whole text is the same idea

Response: Thank you for your comment. This has been changed.

Comment 19: Lines 284-286 the idea requires a bibliographic citation

Response: Thank you for your comment. This has been addded.

Comment 20: Lines 286-287 the idea requires a bibliographic citation

Response: Thank you for your comment. This has been changed.

Comment 21: Lines 288-289 the idea requires a bibliographic citation

Response: Thank you for your comment. This has been added.

Comment 22: Line 289 the idea requires a bibliographic citation

Response: Thank you for your comment. This has been added.

Comment 23: Lines 293-295 the reference [3,4] is not placed in the correct place, I

suggest placing it in (…endocarditis [3,4], including our patient.)

Response: Thank you for your comment. This has been changed.

Comment 24: Lines 358-370 it does not seem to me that the marked text is a summary

Response: Summary statement has been clarified with addition of glomerulonephritis to the complications listed.

Reviewer 3 Report

Comments and Suggestions for Authors

This manuscript is centered on the presentation of a really interesting clinical case, that is of a patient whitout any severe past medical history, who passed over due to an infection caused by an unusual microorganism. 

This case represents a rare clinical presentation of a combination of Endocarditis, accompanied by Leukocytoclastic Vasculitis, pulmonary embolization and Glomerulonephritis. Another point of interest is that Endocarditis presenting with cutaneous vasculitis has been reported previously, altough it is uncommon, which was the case to this patient. Moreover, there are several clinical and laboratory parameters that are uncommonly encounteredin such patients, as they are mentioned in the manuscript (i.e. the patient was the only case that required hemodialysis, only this patient had reported cryoglobulinemia, the immunopathogenesis of glomerulonephritis is different in our case as compared to the prior cases, the abscence of fever, or of a a history of injection drug use and the low rate of endocarditis).

The conclusion section is meaningful and adequatelu supported by the discussion section and the presentation of the patient's case.

Author Response

Reviewer 3 Comments:

This manuscript is centered on the presentation of a really interesting clinical case, that is of a patient whitout any severe past medical history, who passed over due to an infection caused by an unusual microorganism. 

This case represents a rare clinical presentation of a combination of Endocarditis, accompanied by Leukocytoclastic Vasculitis, pulmonary embolization and Glomerulonephritis. Another point of interest is that Endocarditis presenting with cutaneous vasculitis has been reported previously, altough it is uncommon, which was the case to this patient. Moreover, there are several clinical and laboratory parameters that are uncommonly encountered in such patients, as they are mentioned in the manuscript (i.e. the patient was the only case that required hemodialysis, only this patient had reported cryoglobulinemia, the immunopathogenesis of glomerulonephritis is different in our case as compared to the prior cases, the abscence of fever, or of a a history of injection drug use and the low rate of endocarditis).

The conclusion section is meaningful and adequatelu supported by the discussion section and the presentation of the patient's case.

Response: Thank you for taking the time to review our manuscript and provide us with feedback.  We greatly appreciate it!
